# Critical PDT Theory III: Events at the Molecular and Cellular Level

**DOI:** 10.3390/ijms23116195

**Published:** 2022-05-31

**Authors:** David Kessel

**Affiliations:** Department of Pharmacology, Wayne State University School of Medicine, Detroit, MI 48201, USA; dhkessel@med.wayne.edu

**Keywords:** photodynamic therapy, apoptosis, dosimetry

## Abstract

Photodynamic therapy (PDT) is capable of eradicating neoplastic cells that are accessible to sufficient light and oxygen. There is adequate information now available for assessing conditions where PDT might be the therapy of choice, but limited access to clinical facilities and impediments to regulatory approval of new agents have limited clinical usage. Early reports mainly involved clinical data with few thoughts towards finding death pathways. In 2022, there is a clear understanding of the determinants of successful tumor eradication. While PDT may be the optimal method for many clinical indications, support for this approach has lagged. This report provides a commentary on some elements of recent progress in PDT at the molecular and cellular levels, along with a discussion of some of the limitations in current research efforts.

## 1. Introduction

When the concept of photodynamic therapy was being developed, it was not at all clear what this procedure might be able to accomplish. It had been shown in the early part of the 20th century that the combination of light and certain dyes could inactivate microorganisms [1]. Later work established that treatment of cancer patients with a ‘hematoporphyrin derivative’ resulted in malignant lesions becoming fluorescent [2]. While this could be an aid in diagnosis, it was not until Dougherty began his studies that serious efforts were made to promote utilization of what came to be called photodynamic therapy (PDT) for use in the treatment of cancer [3,4].

Dougherty was originally seeking a radiation sensitizer and had asked for advice from Dr. Samuel Schwartz, who had prepared one of the early photosensitizing agents (see below). Schwartz indicated that it might be feasible to use his product for this purpose. He also mentioned the need to keep treated patients away from bright light since their skin would become photosensitized. Dougherty decided at that point to abandon the search for enhancing response to ionizing radiation and turn instead to exploring the use of PDT for cancer treatment. He has provided a summary of this history [5].

## 2. Photodynamic Therapy

During the 1980s, Dougherty organized a series of conferences that involved groups exploring the potential use of PDT for cancer control. It soon became apparent that there were many obstacles to be overcome. The formulation of the photosensitizing agent was highly variable and light sources often involved high-intensity lamps with no controls on wavelength. Clinical protocols varied with the investigator. Nothing was approved by any regulatory agency. These conferences eventually involved workers in the US, Asia and Europe. The implications of PDT were initially unclear. There was no standardization with regard to light sources. The only drug available was the crude mixture that had been prepared by Schwartz and Lipson from hematoporphyrin. 

This agent had an unusual history. In an attempt to purify hematoporphyrin (HP) and provide a water-soluble product, Schwartz first dissolved HP in sulfuric acid, neutralized the resulting solution with base and collected the precipitate. Not knowing what it contained, he named this product hematoporphyrin derivative (HPD). Later work, mainly by Pandey’s group, established that HPD was not a single ‘derivative’, but consisted of a large collection of hematoporphyrin dimers and higher oligomers. These tended to be connected by ether linkages, but some ester linkages were also present. There was also some starting material and other reduced porphyrins [6]. HPD was eventually approved by the FDA for human use in spite of this very complex composition.

Dougherty’s group at the Roswell Park Cancer Center (Buffalo, NY) began treating patients, as did many other groups worldwide. At that point, there was no clear knowledge of the mechanism of action or how PDT should be used. It was later found that both direct tumor photokilling and a shut-down of the tumor vasculature was involved [7,8]. Light sources were eventually developed that could operate at 630 nm, the longest wavelength absorbance optimum for HPD. There were experiments with a variety of systems until laser systems were developed that, coupled with fiber optics, could provide radiation to tumor loci. While a ‘gold-vapor’ laser system could provide light at 630 nm, the usual laser-associated systems involved use of an argon laser coupled with a dye that could transform the photons to 630 nm. Early experiments showed that PDT was best reserved for early tumors of limited size. During conferences in the 1980-90 era, clinical groups would report that attempts at photosensitization and irradiation of large tumors could lead to massive necrotic effects and damage to major blood vessels. These remained mainly unpublished; reports and reviews were confined to favorable indications [9,10]. 

## 3. Photokilling Mechanisms

It is interesting to note that none of the early publications relating to PDT made any reference to mechanisms whereby HPD with light resulted in what has come to be called ‘photokilling’. Reports were mainly concerned with results, with each group having its own protocol. Concepts relating to dosimetry were left for future investigators to solve. 

The first clue concerning mechanisms that could result in photokilling was provided by Oleinick’s group (in 1991). They determined that the lethal effects of PDT resulted from the initiation of apoptosis, an irreversible death pathway [11]. The route to apoptosis was eventually traced to the localization of HPD in mitochondria. Photodamage led to translocation of mitochondrial cytochrome c into the cytoplasm [12], which acted as a trigger for apoptosis [13]. It was later reported that another process termed autophagy was a protective cellular response to photodamage. Autophagy resulted in the appearance of a ‘shoulder’ on the dose-response curve, i.e., this was not a ‘straight line’. This deviation is likely explained by the ability of autophagy to target photodamaged sub-cellular organelles for recycling before they could trigger the initiation of apoptosis. Before Oleinick’s work, there were no clues concerning the mode of action whereby PDT could cause tumor cells to initiate death pathways. 

An additional factor is the ability of PDT to target the tumor vasculature: blood vessels evoked by growth of malignant cells and tissues. This is a well-recognized element of PDT efficacy, although it is poorly understood. There is some intrinsic property of the tumor vasculature that leads to accumulation of photosensitizing agents. The intrinsic properties of this vasculature result in a selective ability to accumulate photosensitizing agents. Subsequent irradiation then leads to vascular shut-down with a significant anti-tumor effect [7,8]. 

Even now, it is not entirely clear why effective photosensitizing agents tend to accumulate in malignant tissues and associated vasculature. An early hint was provided by Jori’s group who noted that low-density lipoprotein receptors might be involved since components of effective photosensitizers became associated with plasma lipoproteins [14]. Overexpression of LDL receptors has been associated with neoplasia [15]. There is also substantial accumulation of many of these agents in liver, spleen, kidney and certain other host organs [16]. These sites are normally protected from light during irradiation procedures. 

While the combination of HPD and light was effective at cancer control, the process had some drawbacks. There is a transient photosensitization of skin, requiring patients to be protected from strong light for several weeks after HPD administration. Light sources centered at 630 nm do not penetrate very far into tissues because of light-scattering effects [17,18]. These problems were solved when it was realized that HPD is not the only effective photosensitizing agent [19,20,21]. Now, many such agents are available with absorbance profiles into far-red and infra-red. This greatly promotes the depth of tissue that can be irradiated. Many of the newer agents do not accumulate in skin, eliminating another PDT adverse effect.

## 4. Recent Research Efforts

The results of continued research into PDT efficacy have resulted in production of more and better photosensitizers, better light sources and an appreciation for dosimetry considerations. There has been substantial progress into exploring the mode of action involved in photokilling. Targeting both mitochondria and lysosomes has been shown to substantially improve photokilling by enhancing the pro-apoptotic effects of photodamage [22]. Newer formulation procedures have been developed that can more specifically target sub-cellular loci [23,24]. Procedures that use combinations of PDT with chemotherapy are being developed. However, along with these improvements in PDT, procedures have come a collection of distractions [25].

Photosensitizing agents are being identified with considerable frequency. Since the cost of obtained regulatory system approval for human use is substantial, it is unlikely that more than a few of these will ever see what is colloquially referred to as ‘the light of day’. Any agent without substantial absorbance at wavelengths of 630 nm or (preferably) above will likely be of little value except for rare cases, e.g., for treatment of bladder cancer [26]. The ability to eradicate tumor cells in monolayers has been demonstrated many times and is only marginally related to problems faced when these results are translated to clinical protocols. Most monolayer studies that use pertinent wavelengths of light show a 90–99% level of tumor eradication with light doses in the 100–500 mJ/cm^2^ range. If a study requires an order of magnitude of greater light dose, something is wrong; either the wavelength is not optimal, the formation of reactive species is insufficient, or the sub-cellular site(s) being targeted are relatively ineffective at initiating lethal effects.

A recent example of translational issues can be found in a recent report in Nature Communications [27]. A series of photosensitizing agents that appear to localize in the golgi was shown to initiate apoptosis upon irradiation at 532 nm. This was effective in both tumor monolayers and in small murine tumors but would clearly be inadequate for treating anything larger. The penetration of green light into tissues is clearly sub-optimal [28]. Such reports generally involve potentially useful elements of photochemistry and photobiology and may provide a basis for further drug development if the absorbance profile of the photosensitizing agents can be altered. 

A convenient method for identifying formation of reactive oxygen species involves use of fluorogenic probes [29,30]. Since these species have very short half-lives, it is obviously necessary for the probe to be there during irradiation. Otherwise, only a few long-lived species will be detected. It is not uncommon to see reports where the probe is added after irradiation [25] where only a few long-lived species would be detected, e.g., lipid peroxides. 

An important element in the efficacy of PDT is the pattern of sub-cellular localization of photosensitizing agents. This can vary widely, as is shown in Figure 1. An early study had shown that a two-photosensitizer combination that targeted mitochondria and lysosomes could significantly promote eradication of large tumors in the rat [31]. It was initially thought that this effect resulted from the simultaneous targeting of both tumor and vasculature for photodamage. Later studies revealed that the observed synergism was the result of targeting of both lysosomes and mitochondria [22]. The pertinent clue was that the order of activation of the sensitizers was a critical factor. Lysosomal photodamage was found to initiate an effect that promoted the apoptotic pathway initiated by mitochondrial perturbation [22,32]. A summary of the pathways involved has been provided [32]. Lysosomal photodamage promotes loss of lysosomal calcium which, in turn, activates the protease calpain. This can also mediate the cleavage of the apoptosis-associated protein ATG5 to a fragment termed tATG5 that attaches to the mitochondrial membrane and promotes a sequence of events that ultimately leads to loss of cytochrome c, a trigger for apoptosis [33]. Since tATG5 has a very short half-life and is readily degraded by stray proteases, it is important that this fragment be released after the initial photodamage to mitochondria has occurred. This accounts for the requirement, demonstrated in Ref. [31], that mitochondrial damage must occur first for an optimal effect.

Use of this dual targeting approach resulted in a substantial increase in photokilling of cells in 3-D culture [34]. Data are summarized in Figure 2. This is an important consideration since the effect did not involve increasing the light dose, but rather was involved in an increase in the efficacy of a given light dose. In studies summarized in Ref. [34], a single photosensitizer was used. This agent, benzoporphyrin BPD (benzoporphyrin derivative, Visudyne) normally targets mitochondria and the ER. A nano-formulation was designed that can redirect this agent to lysosomes [34]. Using a single agent and a single wavelength of light, it was, therefore, possible to provide an optimal selection of subcellular sites for photodamage. Simultaneous targeting of mitochondria and lysosomes was therefore feasible. Otherwise, it is necessary to use two different photosensitizers and two different wavelengths of light [31] which complicates the protocol by adding this additional factor along with the need to consider pharmacokinetics of two different agents. 

An additional advantage to the use of BPD as a photosensitizing agent relates to its targeting profile. Among the targets for photosensitization is the ER. A sufficient level of ER stress can lead to a novel, and mainly unexplored, death pathway termed paraptosis [35]. During this process, the cytoplasm begins to fill with vacuoles derived from ER structures and is eventually shed into the environment, leaving bare nuclei behind. This pathway to cell death is independent of apoptosis and can lead to the death of cells even where the pathway to apoptosis is impaired [36]. A comparison of the morphology of apoptosis and paraptosis is shown in Figure 3. The ability of nano-formulation procedures to redirect sub-cellular localization is an example of the progress in PDT that can promote efficacy of agents already shown to be safe and effective. Paraptosis has yet to be more thoroughly explored as a potential factor in photokilling by PDT. In theory, any agent that perturbs ER should be capable of inducing paraptosis: this would include BPD, hypericin, m-THPC and many other agents. Unfortunately, most investigators do not examine cells after photodamage using phase-contrast microscopy (see Figure 3). The morphology of paraptosis can readily be distinguished from apoptosis. 

An important aspect of current PDT research involves exploration of a variety of methods for promoting tumor eradication by what has been termed ‘deep-tissue activation’, i.e., dealing with neoplasia that has a substantial bulk. A recent review on this topic has appeared [38]. Procedures suggested include the use of two-photon irradiation systems, Cerenkov radiation and use of ultrasound or ionizing radiation in conjunction with photo-nanomedicines. In this regard, it is important to recall some of the issues encountered in the early days of PDT that have not been thoroughly reported in the literature. When tumors become large, they need to evoke a blood supply in order to survive. Attempts at ‘deep’ tumor eradication could result in damage to large blood vessels and serious consequences for the patient. A better approach, described in the context of treatment involving head and neck cancer, involves prior surgical debulking [39]. This minimizes the extent of neoplasia being treated and the likelihood of blood-vessel damage. PDT is much more successful when used in situations where major blood vessels are not in the light path. Use of Cerenkov radiation to trigger photodynamic effects involves use of ionizing radiation to initiate the production of visible photons [40]. Even at a 2 Gy radiation level, a typical visible light emission might be of the order of 0.1 mJ/cm^2^, which is several orders of magnitude less light than is required for an adequate PDT light dose. The latter case is an example of a research direction that appears to be feasible until one considers the limitations that will be encountered in practice. 

## 5. Conclusions

This is the third in a series of reports on critical appraisals of PDT research. Other discussions will soon appear in *Photochemistry and Photobiology* and in *Photodiagnosis and Photodynamic Therapy*. Successful application of PDT can lead to the eradication of localized neoplasia without many of the adverse effects associated with chemotherapy and ionizing radiation. Protocols involving combinations of PDT and chemotherapy are also being investigated [41,42,43]. The news that yet one more product can catalyze formation of reactive species upon irradiation is of limited interest unless that agent has some unusual properties that are superior to agents already available. Many such reports do not advance the field and often describe agents that would never be clinically useful. Any agent with no significant absorbance above 600 nm is unlikely to be of use in clinical PDT, except perhaps in very special cases. PDT does offer an approach to treating localized lesions that may be resistant to conventional chemotherapy and/or radiation. Critical factors with regard to potential clinical relevance will be ease of formulation, absorbance spectra, sub-cellular localization phenomena, efficiency in catalyzing formation of reactive oxygen species and potential for also initiating vascular shutdown. This report has not discussed the implications of vascular effects in PDT, which are substantial and are discussed in detail elsewhere [44,45]. 

Those involved in drug development in the realm of PDT need to consider the same criteria that are involved in the evaluation of chemotherapy. The first ‘phase’ of any successful proposal consists of toxicity studies. Agents that initiate adverse effects are unlikely to advance. The second step is to discover the spectrum of efficacy: for which indications is this therapy likely to be helpful? The final step involves comparison with current therapy: does a new agent show any advantages over current procedures? With a substantial collection of photosensitizing agents currently in use, it will be necessary to determine whether any new agents represent a significant improvement, since achieving regulatory approval is an expensive and lengthy process.

In this report, I have mainly dealt with issues relating to photodynamic therapy at the cellular and molecular levels. For readers unfamiliar with the topic, there are numerous reviews on this topic and a recent compendium of all reviews has been published [46]. 

## Figures and Tables

**Figure 1 ijms-23-06195-f001:**
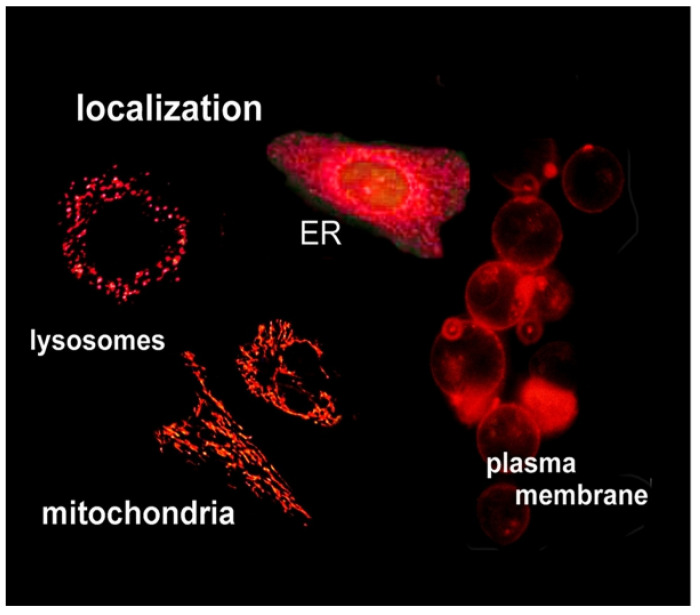
Fluorescence microscopy studies showing the ability of this technique to delineate the preferential localization of photosensitizing agents.

**Figure 2 ijms-23-06195-f002:**
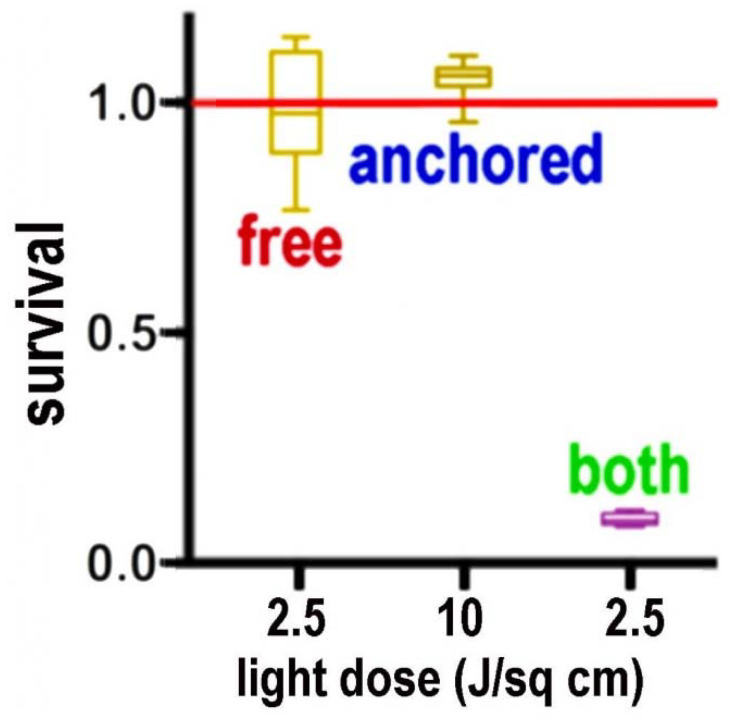
Photokilling in 3D culture showing the synergistic effect of simultaneously targeting mitochondria and ER with benzoporphyrin derivative (BPD). Free BPD targets mitochondria > ER; ‘anchored’ BPD targets lysosomes. See Ref. [33] for further details.

**Figure 3 ijms-23-06195-f003:**
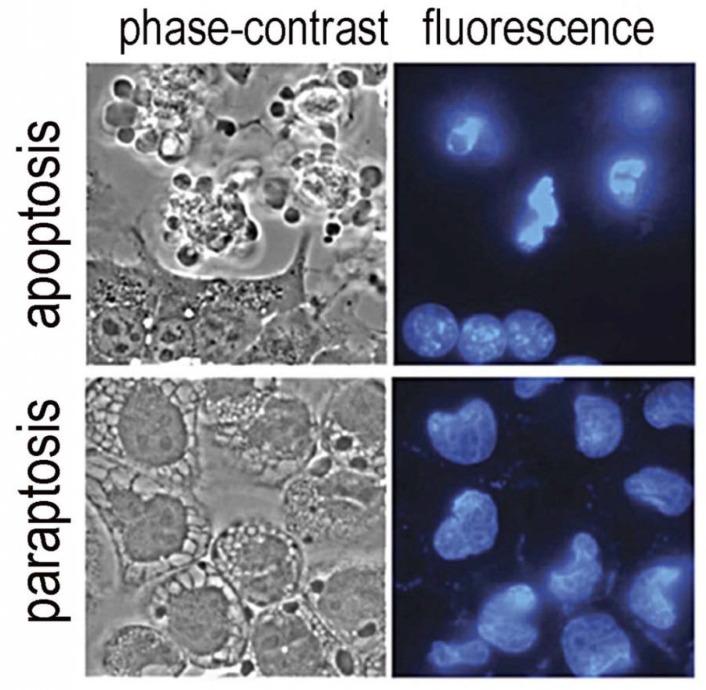
Morphology of apoptosis vs. paraptosis. Phase contrast images are compared with fluorescence-labeling of nuclei with Höchst 33342 [37].

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
