# Peer review of "Critical PDT Theory III: Events at the Molecular and Cellular Level"

_ijms, 2022, doi:10.3390/ijms23116195_

Round 1

Reviewer 1 Report

General remarks:

-- the title of the paper is a bit misleading. At first I assumed that "Let there be light" would be a review on illumination techniques, light sources, light delivery schemes and dosimetry. However, these topics are covered only partially and it is not the main focus. A second guess was that "Let there be light" refers to the historical origins of PDT. However, this is again covered only partially and it is not the main focus. I suggest to change the title so that it matches better the main focus of the paper. This will help reader decide whether the work is relevant for him.

-- While the paper provides and interesting personal insights into several aspects of PDT, it lacks a clear focus. There is a brief history of PDT, several thoughts on absorption profile of photosensitizers; a quite detailed discussion of dual targeting of BPD into mitochondria and lysosomes and the resulting synergic effect; presentation of paraptosis as a not so widely known cell death pathway and then several thoughts on dead-ends and 'useless' research in PDT, where the author points to research that is unlikely to ever be translated into clinical praxis. While these are interesting points, it is not clear whether these are the most important topics that widely resonate in the PDT research community or whether this is rather author's selection that fits into his personal research niche. As the paper aims to be a review and hence a possible entry point into the topic for a newcomer, I recommend to cover at least briefly the main principles of PDT, current clinical status and the main current research efforts and trends. Some of this may be the focus of another part of the series (which I do not have access to), but it should be clear from the title and abstract what is the focus of this part of the series.

-- There are two sensitizers that are discussed in the paper (HPD and BPD). I think that the author should review briefly what are other clinically approved and sensitizers and discuss the span of their use.

-- The author could briefly review for which diseases and indications PDT is used and critically evaluate whether the use of PDT is on the rise or not (and for which indications). This would help to put the other points into context.

-- The author had several thoughts on development of new sensitizers and seemed to be rather skeptical, perhaps expressing a feeling that more effort should be put into clinical translation of existing photosensitizers rather than developing new ones(?). This topic could be handled in more detail: Which of the approved sensitizers are the most promising for new indications? Which sensitizers that are in research stage are the most promising and deserve to be evaluated in a larger clinical trial?

-- I suggest to add more figures, for example a graph of tissue absorption/scattering spectrum, or figure with structures and/or absorption spectra of the most widely used sensitizers.

-- There are 12 citations to works from the past 10 years, out of which 6 are autocitations. 

-----------------------

Specific remarks, typos:

-- Line 12: "lagged, This": comma instead of a dot

-- Line 28: "sources.the": redundant dot

-- Line 42: unfinished phrase

-- Line 60: "dye dye": doubled word

-- Line 77: 'shoulder' on a dose response curve: It is not clear to me what this means

-- Line 124: "500 mJ/sq cm": standard notation is 500 mJ/cm2 with a superscript. This is repeated several times.

-- Line 128: "An recent"

-- Line 143: "This can very widely": very--> vary

-- Line 167: redundant "which"

-- Line 193: "may"-->"many"?

-- Figure 1 caption: "agents:."

-- Figure 1 caption: own work/adopted/ref?

-- Figure 3 caption: own work/adopted/ref?

Author Response

With regard to the commentary of  Reviewer 1: the title has been altered to reflect what is intended. This is not meant to be a review of PDT since there are many now available. Instead, this is intended to be a critical discussion of what I consider to be the relevant vs. irrelevant issues in some of the current research efforts. The intent is not to describe what use can be made of PDT, or what indications show promise for photodynamic treatment. 

Assorted errors in the text have been corrected. I believe that readers will understand the meaning of mJ/sq cm. Figures 1 and 3 were created for this report from prior data.

Reviewer 2 Report

This is an outstanding review of the mechanisms involved in PDT at the molecular and cellular level by a recognised and highly respected global expert in the field. However, the content does not fit well with the title, unless this is identified as one of a series of papers on different aspects of PDT, which may indeed be the case as the title is Critical Theory III. There is reference to another Critical Theory paper (no number) but no mention of papers I and II. Further, the detailed scientific material does not match the abstract well. The abstract implies clinical content, whereas covering of the key clinical aspects is superficial in the text compared with the high sophistication of the science at the cellular and molecular level.  There is a considerable gap between the data presented here, what is implied in the abstract, and potential clinical applications.

A more suitable title would be PDT at the molecular and cellular level 

To be clinically useful, the macroscopic effects of PDT and the nature of the tissues left behind after healing must be understood in both normal and diseased tissues. The selectivity of PDT between normal and diseased tissue is relative, not absolute, which is particularly important in sites where tumours meet normal tissue. One of the beauties is that so many normal tissues heal after PDT without any unacceptable effects on the structure or function of those organs, but that does not necessarily mean that there is no effect. Further, there is plenty of room for trouble if effects on normal tissue are not properly understood. This happened in the early days of PDT with the disaster of shrivelled scarred contracted bladders after treatment of carcinoma in situ of the bladder mucosa. This happened because nobody had taken the trouble to see what PDT did to normal parts of the bladder, such as the submucosal muscle, prior to jumping into treating patients.

I am concerned about the comments made about effects on arteries. Very small vessels (<0.5mm) with little connective tissue in the vessel wall feeding cancers can be destroyed by PDT and this plays an important part in therapy. In normal larger arteries, PDT destroys the endothelium and smooth muscle in the arterial walls, but without causing rupture or stenosis as the connective tissue in the arterial wall is preserved. The endothelium regenerates within a week or so. This effect on the smooth muscle can reduce the risk of intimal hyperplasia after balloon angioplasty and in some cases may reduce the need for post angioplasty stenting, so this has considerable potential clinical benefits. On the other hand, tumour invasion of arterial wall can deprive the vessel of its mechanical strength, so may lead to rupture after PDT. This is acknowledged in the text (line 211) by saying that it is best not to treat tumours adjacent to large arteries, but no explanation is given.

A reference to bulk tumour eradication is given, but very superficially. Image guided PDT of larger lesions is now very sophisticated, but little relevant information on this is given in the text

Most of the conclusions given are very sound, but the list of relevant factors for improvement is limited to effects at the cellular level. There is no mention of crucial factors like what tissues will be left after PDT, whether mechanical strength or function will be preserved or damaged etc, etc.

 Note : Abstract line 42: incomplete sentence

Author Response

Reviewer 2 appears to have grasped the point and also suggests amending the title which has been done. My intent in this report has been to provide a brief background so as to put into perspective critical remarks about some current PDT reports that are not advancing the field. I have added references to papers I and II in this series. Since effects of PDT on major blood vessels is a critical issue, I have added a bit more on the subject in the conclusion section, but this is not a major aspect of this report. I have mainly considered cellular rather than more global effects of PDT in this report. Since this reviewer clearly has a more pertinent clinical outlook, I encourage the preparation of a similar report discussing these factors.

Round 2

Reviewer 1 Report

The author has addressed a larger part of my comments and suggestions. The title has been changed to a more appropriate one and the abstract has been altered too to better reflect the content of the paper. Typos and minor errors that I pointed out have been corrected. The author clarified the focus and purpose of the paper in the cover letter.

I like the idea to review trials and errors, pointing out some efforts that were particularly fruitful and on the other hand those that are unlikely to advance the field. However, I think that this could be done in a bit more systematic way. Moreover, some of the major concerns that I raised in the first review have not been addressed.   

-- I think that it is desirable to add a brief review (as a separate paragraph/section) of current clinical status of photosensitizers, the list of approved indications and information about their absorption profiles, absorption coefficients, subcellular localization etc., because these aspects are further discussed in the paper and adding a new section would help the reader to put the information into context and broader perspective.  

-- The selection of topics that are commented on in the paper seems to be a bit random. There are several comments on i) some absorption profiles of new photosensitizers, ii) an isolated comment on wrong approach to detect ROS, iii) subcellular localization and dual targeting, iv) cell death modes and paraptosis. First, I think that the comments on light absorption and light dose could be put together. The last paragraph before Conclusions dealing with Cerenkov could be moved so that it is together with other comments on light doses. Second, the topics iii) and iv) are covered more in detail as these probably fall into author's realm of interest and point to his own research. It is not clear whether these are among the most important breakthroughs in the field or whether they have been selected because these topics are author's interests. A bit more perspective should be provided to the reader about selection of the topics.  

-- The unit 'J/sq cm' is not a standard notation and I do not see a reason why not to use the standard "J/cm^2".

Author Response

Since the major thrust of the article does relate to the second choice, I have no objection to ‘Critical PDT Theory III: events at the molecular and cellular level’.

 Photodynamic therapy (PDT) is capable of eradicating neoplastic cells accessible to sufficient light and oxygen. There is adequate information for assessing conditions where PDT might be the therapy of choice. Early reports mainly involved clinical data with few thoughts toward finding death pathways. In 2022, there is a clear understanding of the determinants of successful tumor eradication. While PDT may be the optimal method for many clinical indications, support for this approach has lagged. This report provides a commentary on some elements of recent progress in PDT at the molecular and cellular level, along with a discussion of some of the limitations in current research efforts. 

In this report, I comment on methodology relating to the ‘molecular and cellular level, not clinical trials and regulatory approval. Those issues relate to the accrual of clinical data and studies on toxicity, appropriate indications for PDT and whether this form of therapy represents an advance in patient care and are beyond the scope of this report. 

This was never intended to be a discussion of the clinical status of PDT. This is best left to those who deal with clinical aspects of cancer. Some of the comments have been rearranged as suggested. I have added a few thoughts on the disconnect between what might be termed ‘basic research’ and ‘regulatory approval’.  This was not intended to be a review of PDT, of which there are many currently in print. I have added a reference to a recent ‘compendium’ of all recent reviews. Since the PDT literature tends to use the J/cm^2 format for discussing light doses, I can change the current terminology to this option. As for further rearranging the test, my impression remains that the topics have been dealt with in the appropriate order.  

Reviewer 2 Report

The author has partly responded to my comments. However, I feel that the new title is no better than the first one. This paper is not about trials and errors. It is all about reviewing the current understanding of the mechanisms of PDT at the molecular and cellular level. I appreciate this is one paper in a series of articles for a focused journal issue and it may be appropriate to make the title of this article complement others in the issue but it would seem reasonable for it to have at least some direct relevance to the content.   Perhaps:

 Critical PDT Theory – what happens under the microscope,     or

Critical PDT Theory at the molecular and cellular level

I also feel that the abstract still does not appropriately reflect the scientific content of the main text.  I would suggest revisions in the abstract along the lines of:

a)     There is adequate information now available for assessing conditions
where PDT might be the therapy of choice, but limited undertaking of the clinical trials that are required for regulatory approval ……

b)     this report provides commentary on recent progress in PDT at the molecular and cellular level….

The other minor edits in the text are all fine

Author Response

I have altered the title and expanded the Abstract as recommended. Some additional changes were made in response to the other reviewer.